# Generalized Sliced Wasserstein Distances

**Soheil Kolouri**[1]*, **Kimia Nadjahi**[2]*, **Umut Şimşekli**[2,3], **Roland Badeau**[2], **Gustavo K. Rohde**[4]

1: HRL Laboratories, LLC., Malibu, CA, USA, 90265
2: LTCI, Télécom Paris, Institut Polytechnique de Paris, France
3: Department of Statistics, University of Oxford, UK
4: University of Virginia, Charlottesville, VA, USA, 22904
`skolouri@hrl.com, gustavo@virginia.edu`
`{kimia.nadjahi, umut.simsekli, roland.badeau}@telecom-paris.fr`

## Abstract

The Wasserstein distance and its variations, e.g., the sliced-Wasserstein (SW) distance, have recently drawn attention from the machine learning community. The SW distance, specifically, was shown to have similar properties to the Wasserstein distance, while being much simpler to compute, and is therefore used in various applications including generative modeling and general supervised/unsupervised learning. In this paper, we first clarify the mathematical connection between the SW distance and the Radon transform. We then utilize the generalized Radon transform to define a new family of distances for probability measures, which we call generalized sliced-Wasserstein (GSW) distances. We further show that, similar to the SW distance, the GSW distance can be extended to a maximum GSW (max-GSW) distance. We then provide the conditions under which GSW and max-GSW distances are indeed proper metrics. Finally, we compare the numerical performance of the proposed distances on the generative modeling task of SW flows and report favorable results.

## 1 Introduction

The Wasserstein distance has its roots in optimal transport (OT) theory [1] and forms a metric between two probability measures. It has attracted abundant attention in data sciences and machine learning due to its convenient theoretical properties and applications on many domains [2, 3, 4, 5, 6, 7, 8], especially in implicit generative modeling such as OT-based generative adversarial networks (GANs) and variational auto-encoders [9, 10, 11, 12].

While OT brings new perspectives and principled ways to formalize problems, the OT-based methods usually suffer from high computational complexity. The Wasserstein distance is often the computational bottleneck and it turns out that evaluating it between multi-dimensional measures is numerically intractable in general. This important computational burden is a major limiting factor in the application of OT distances to large-scale data analysis. Recently, several numerical methods have been proposed to speed-up the evaluation of the Wasserstein distance. For instance, entropic regularization techniques [13, 14, 15] provide a fast approximation to the Wasserstein distance by regularizing the original OT problem with an entropy term. The linear OT approach, [16, 17] further simplifies this computation for a given dataset by a linear approximation of pairwise distances with a functional defined on distances to a reference measure. Other notable contributions towards computational methods for OT include multi-scale and sparse approximation approaches [18, 19], and Newton-based schemes for semi-discrete OT [20, 21].

There are some special favorable cases where solving the OT problem is easy and reasonably cheap. In particular, the Wasserstein distance for one-dimensional probability densities has a closed-

form formula that can be efficiently approximated. This nice property motivates the use of the sliced-Wasserstein distance [22], an alternative OT distance obtained by computing infinitely many *linear projections* of the high-dimensional distribution to one-dimensional distributions and then computing the average of the Wasserstein distance between these one-dimensional representations. While having similar theoretical properties [23], the sliced-Wasserstein distance has significantly lower computational requirements than the classical Wasserstein distance. Therefore, it has recently attracted ample attention and successfully been applied to a variety of practical tasks [22, 24, 25, 26, 27, 28, 29, 30, 31].

As we will detail in the next sections, the linear projection process used in the sliced-Wasserstein distance is closely related to the Radon transform, which is widely used in tomography [32, 33]. In other words, the sliced-Wasserstein distance is calculated via linear slicing of the probability distributions. However, the linear nature of these projections does not guarantee an efficient evaluation of the sliced-Wasserstein distance: in very high-dimensional settings, the data often lives in a thin manifold and the number of randomly chosen linear projections required to capture the structure of the data distribution grows very quickly [27]. Reducing the number of required projections would thus result in a significant performance improvement in sliced-Wasserstein computations.

To address the inefficiencies caused by the linear projections, very recently, several attempts have been made. In [34], Rowland et al. combined linear projections with orthogonal coupling in Monte Carlo estimation to increase computational efficiency and estimation quality. In [35], Deshpande et al. extended the sliced-Wasserstein distance to the 'max-sliced-Wasserstein' distance, where they aimed at finding a single linear projection that maximizes the distance in the projected space. In another study [36], Paty and Cuturi extended this idea to projection on *linear subspaces*, where they aimed at finding the optimal subspace for the projections by replacing the projections along a vector with projections onto the nullspace of a matrix. While these methods reduce the computational cost induced by the projection operations by choosing a single vector or an orthogonal matrix, they incur an additional cost since they need to solve a non-convex optimization over manifolds.

In this paper, we address the aforementioned computational issues of the sliced-Wasserstein distance by taking an alternative route. In particular, we extend the linear slicing to *non-linear* slicing of probability measures. Our main contributions are summarized as follows:

- Using the theory of the *generalized* Radon transform [37] we extend the definition of the sliced-Wasserstein distance to an entire class of distances, which we call the generalized sliced-Wasserstein (GSW) distance. We prove that replacing the linear projections with *non-linear* projections can still yield a valid distance metric and we then identify general conditions under which the GSW distance is a proper metric function. To the best of our knowledge, this is the first study to generalize SW to non-linear projection.
- Similar to [35], we then show that, instead of using infinitely many projections as required by the GSW distance, we can still define a valid distance metric by using a *single* projection, as long as the projection gives the maximal distance in the projected space. We aptly call this distance the max-GSW distance.
- As instances of non-linear projections, we first investigate projections with *polynomial* kernels, which fulfill all the conditions that we identify. However, we observe that the memory complexity of such projections has a combinatorial growth with respect to the dimension of the problem, hence restricts their applications to modern problems. This motivates us to consider a neural-network-based projection scheme, where we observe that fully connected or convolutional networks with leaky ReLU activations fulfill all the crucial conditions so that their resulting GSW becomes a pseudo-metric for probability measures. In addition to its practical advantages, this scheme also brings an interesting perspective on adversarial generative modeling, showing that such algorithms contain an implicit stage for learning projections with different cost functions than ours.
- Due to their inherent non-linearity, the GSW and max-GSW distances are expected to capture the complex structure of high-dimensional distributions by using much less projections, which will reduce the iteration complexity in a significant amount. We verify this fact in our experiments, where we illustrate the superior performance of the proposed distances in both synthetic and real-data settings.

## 2 Background

We review in this section the preliminary concepts and formulations needed to develop our framework, namely the $p$-Wasserstein distance, the Radon transform, the sliced $p$-Wasserstein distance and the maximum sliced $p$-Wasserstein distance. In what follows, we denote by $P_p(\Omega)$ the set of Borel probability measures with finite $p$'th moment defined on a given metric space $(\Omega, d)$ and by $\mu \in P_p(X)$ and $\nu \in P_p(Y)$ probability measures defined on $X, Y \subseteq \Omega$ with corresponding probability density functions $I_\mu$ and $I_\nu$, i.e. $d\mu(x) = I_\mu(x)dx$ and $d\nu(y) = I_\nu(y)dy$.

**Wasserstein Distance.** The $p$-Wasserstein distance, $p \in [1, \infty)$, between $\mu$ and $\nu$ is defined as the solution of the optimal mass transportation problem [1]:

$$W_p(\mu, \nu) = \left( \inf_{\gamma \in \Gamma(\mu,\nu)} \int_{X \times Y} d^p(x, y) d\gamma(x, y) \right)^{\frac{1}{p}} \tag{1}$$

where $d^p(\cdot, \cdot)$ is the cost function, and $\Gamma(\mu, \nu)$ is the set of all transportation plans $\gamma \in \Gamma(\mu, \nu)$ such that:

$$\gamma(A \times Y) = \mu(A) \text{ for any Borel } A \subseteq X, \quad \gamma(X \times B) = \nu(B) \text{ for any Borel } B \subseteq Y.$$

Due to Brenier's theorem [38], for absolutely continuous probability measures $\mu$ and $\nu$ (with respect to the Lebesgue measure), the $p$-Wasserstein distance can be equivalently obtained from

$$W_p(\mu, \nu) = \left( \inf_{f \in MP(\mu,\nu)} \int_X d^p(x, f(x)) d\mu(x) \right)^{\frac{1}{p}} \tag{2}$$

where $MP(\mu, \nu) = \{f : X \to Y \mid f_\# \mu = \nu\}$ and $f_\# \mu$ represents the pushforward of measure $\mu$, characterized as

$$\int_A df_\# \mu(y) = \int_{f^{-1}(A)} d\mu(x) \text{ for any Borel subset } A \subseteq Y.$$

Note that in most engineering and computer science applications, $\Omega$ is a compact subset of $\mathbb{R}^d$ and $d(x, y) = |x - y|$ is the Euclidean distance. By abuse of notation, we will use $W_p(\mu, \nu)$ and $W_p(I_\mu, I_\nu)$ interchangeably.

**One-dimensional distributions:** The case of one-dimensional continuous probability measures is specifically interesting as the $p$-Wasserstein distance has a closed-form solution. More precisely, for one-dimensional probability measures, there exists a unique monotonically increasing transport map that pushes one measure to another. Let $F_\mu(x) = \mu((-\infty, x]) = \int_{-\infty}^x I_\mu(\tau)d\tau$ be the cumulative distribution function (CDF) for $I_\mu$ and define $F_\nu$ to be the CDF of $I_\nu$. The optimal transport map is then uniquely defined as $f(x) = F_\nu^{-1}(F_\mu(x))$ and, consequently, the $p$-Wasserstein distance has an analytical form given as follows:

$$W_p(\mu, \nu) = \left( \int_X d^p(x, F_\nu^{-1}(F_\mu(x))) d\mu(x) \right)^{\frac{1}{p}} = \left( \int_0^1 d^p(F_\mu^{-1}(z), F_\nu^{-1}(z)) dz \right)^{\frac{1}{p}} \tag{3}$$

where Eq. (3) results from the change of variable $F_\mu(x) = z$. Note that for empirical distributions, Eq. (3) is calculated by simply sorting the samples from the two distributions and calculating the average $d^p(\cdot, \cdot)$ between the sorted samples. This requires only $O(M)$ operations at best and $O(M \log M)$ at worst, where $M$ is the number of samples drawn from each distribution (see [30] for more details). The closed-form solution of the $p$-Wasserstein distance for one-dimensional distributions is an attractive property that gives rise to the sliced-Wasserstein (SW) distance. Next, we review the Radon transform, which enables the definition of the SW distance. We also formulate an alternative OT distance called the maximum sliced-Wasserstein distance.

**Radon Transform.** The standard Radon transform, denoted by $\mathcal{R}$, maps a function $I \in L^1(\mathbb{R}^d)$, where

$$L^1(\mathbb{R}^d) = \{I : \mathbb{R}^d \to \mathbb{R} \ / \int_{\mathbb{R}^d} |I(x)| dx < \infty\},$$

to the infinite set of its integrals over the hyperplanes of $\mathbb{R}^d$ and is defined as

$$\mathcal{R}I(t, \theta) = \int_{\mathbb{R}^d} I(x)\delta(t - \langle x, \theta \rangle) dx, \tag{4}$$

for $(t, \theta) \in \mathbb{R} \times \mathbb{S}^{d-1}$, where $\mathbb{S}^{d-1} \subset \mathbb{R}^d$ stands for the $d$-dimensional unit sphere, $\delta(\cdot)$ the one-dimensional Dirac delta function, and $\langle \cdot, \cdot \rangle$ the Euclidean inner-product. Note that $\mathcal{R} : L^1(\mathbb{R}^d) \to L^1(\mathbb{R} \times \mathbb{S}^{d-1})$. Each hyperplane can be written as:

$$H(t, \theta) = \{x \in \mathbb{R}^d \mid \langle x, \theta \rangle = t\}, \tag{5}$$

which alternatively can be interpreted as a level set of the function $g \in \mathbb{R}^d \times \mathbb{S}^{d-1} \to \mathbb{R}$ defined as $g(x, \theta) = \langle x, \theta \rangle$. For a fixed $\theta$, the integrals over all hyperplanes orthogonal to $\theta$ define a continuous function $\mathcal{R}I(\cdot, \theta) : \mathbb{R} \to \mathbb{R}$ which is a projection (or a slice) of $I$.

The Radon transform is a linear bijection [39, 33] and its inverse $\mathcal{R}^{-1}$ is defined as:

$$I(x) = \mathcal{R}^{-1}\big(\mathcal{R}I(t, \theta)\big) = \int_{\mathbb{S}^{d-1}} (\mathcal{R}I(\langle x, \theta \rangle, \theta) * \eta(\langle x, \theta \rangle) d\theta \tag{6}$$

where $\eta(\cdot)$ is a one-dimensional high-pass filter with corresponding Fourier transform $\mathcal{F}\eta(\omega) = c|\omega|^{d-1}$, which appears due to the Fourier slice theorem [33], and '$*$' is the convolution operator. The above definition of the inverse Radon transform is also known as the filtered back-projection method, which is extensively used in image reconstruction in the biomedical imaging community. Intuitively each one-dimensional projection (or slice) $\mathcal{R}I(\cdot, \theta)$ is first filtered via a high-pass filter and then smeared back into $\mathbb{R}^d$ along $H(\cdot, \theta)$ to approximate $I$. The summation of all smeared approximations then reconstructs $I$. Note that in practice, acquiring an infinite number of projections is not feasible, therefore the integration in the filtered back-projection formulation is replaced with a finite summation over projections (*i.e.*, a Monte-Carlo approximation).

**Sliced-Wasserstein and Maximum Sliced-Wasserstein Distances.** The idea behind the sliced $p$-Wasserstein distance is to first, obtain a family of one-dimensional representations for a higher-dimensional probability distribution through linear projections (via the Radon transform), and then, calculate the distance between two input distributions as a functional on the $p$-Wasserstein distance of their one-dimensional representations (*i.e.*, the one-dimensional marginals). The sliced $p$-Wasserstein distance between $I_\mu$ and $I_\nu$ is then formally defined as:

$$SW_p(I_\mu, I_\nu) = \left( \int_{\mathbb{S}^{d-1}} W_p^p\big(\mathcal{R}I_\mu(., \theta), \mathcal{R}I_\nu(., \theta)\big) d\theta \right)^{\frac{1}{p}} \tag{7}$$

This is indeed a distance function as it satisfies positive-definiteness, symmetry and the triangle inequality [23, 24].

The computation of the SW distance requires an integration over the unit sphere in $\mathbb{R}^d$. In practice, this integration is approximated by using a simple Monte Carlo scheme that draws samples $\{\theta_l\}$ from the uniform distribution on $\mathbb{S}^{d-1}$ and replaces the integral with a finite-sample average:

$$SW_p(I_\mu, I_\nu) \approx \left( \frac{1}{L} \sum_{l=1}^{L} W_p^p\big(\mathcal{R}I_\mu(\cdot, \theta_l), \mathcal{R}I_\nu(\cdot, \theta_l)\big) \right)^{1/p} \tag{8}$$

In higher dimensions, the random nature of slices could lead to underestimating the distance between the two probability measures. To further clarify this, let $I_\mu = \mathcal{N}(0, I_d)$ and $I_\nu = \mathcal{N}(x_0, I_d), x_0 \in \mathbb{R}^d$, be two multivariate Gaussian densities with the identity matrix as the covariance matrix. Their projected representations are one-dimensional Gaussian distributions of the form $\mathcal{R}I_\mu(\cdot, \theta) = \mathcal{N}(0, 1)$ and $\mathcal{R}I_\nu(\cdot, \theta) = \mathcal{N}(\langle \theta, x_0 \rangle, 1)$. It is therefore clear that $W_2(\mathcal{R}I_\mu(\cdot, \theta), \mathcal{R}I_\nu(\cdot, \theta))$ achieves its maximum value when $\theta = \frac{x_0}{\|x_0\|_2}$ and is zero for $\theta$'s that are orthogonal to $x_0$. On the other hand, we know that vectors that are randomly picked from the unit sphere are more likely to be nearly orthogonal in high-dimension. More rigorously, the following inequality holds: $Pr\big(|\langle \theta, \frac{x_0}{\|x_0\|_2} \rangle| < \epsilon \big) > 1 - e^{(-d\epsilon^2)}$, which implies that for a high dimension $d$, the majority of sampled $\theta$'s would be nearly orthogonal to $x_0$ and therefore, $W_2(\mathcal{R}I_\mu(\cdot, \theta), \mathcal{R}I_\nu(\cdot, \theta)) \approx 0$ with high probability.

To remedy this issue, one can avoid uniform sampling of the unit sphere, and pick samples $\theta$'s that contain discriminant information between $I_\mu$ and $I_\nu$ instead. This idea was for instance used in [28, 35, 36]. For instance, Deshpande et al. [28] first calculate a linear discriminant subspace and then measure the empirical SW distance by setting the $\theta$'s to be the discriminant components of the subspace.

A similarly flavored but less heuristic approach is to use the maximum sliced $p$-Wasserstein (max-SW) distance, which is an alternative OT metric defined as [35]:

$$\text{max-}SW_p(I_\mu, I_\nu) = \max_{\theta \in \mathbb{S}^{d-1}} W_p\big(\mathcal{R}I_\mu(\cdot, \theta), \mathcal{R}I_\nu(\cdot, \theta)\big) \tag{9}$$

Given that $W_p$ is a distance, it is straightforward to show that max-$SW_p$ is also a distance: we will prove in Section 3.2 that the metric axioms would also hold for the maximum generalized sliced-Wasserstein distance, which contains the max-SW distance as a special case.

# 3 Generalized Sliced-Wasserstein Distances

We propose in this paper to extend the definition of the sliced-Wasserstein distance to formulate a new optimal transport metric, which we call the generalized sliced-Wasserstein (GSW) distance. The GSW distance is obtained using the same procedure as for the SW distance, except that here, the one-dimensional representations are acquired through nonlinear projections. In this section, we first review the generalized Radon transform, which is used to project the high-dimensional distributions, and we then formally define the class of GSW distances. We also extend the concept of max-SW distance to the class of maximum generalized sliced-Wasserstein (max-GSW) distances.

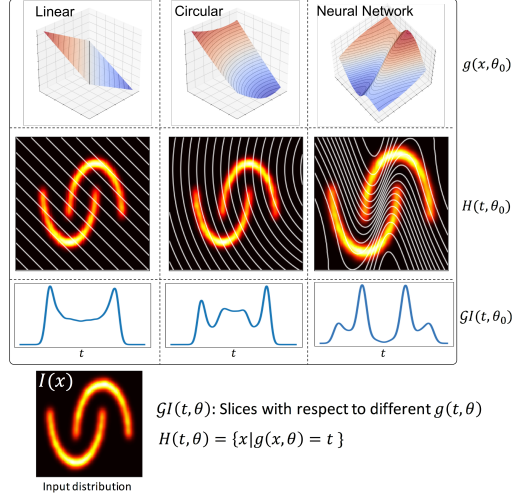

Figure 1: Visualizing the slicing process for classical and generalized Radon transforms for the Half Moons distribution. The slices $\mathcal{G}I(t, \theta)$ follow Equation (10).

## 3.1 Generalized Radon Transform

The generalized Radon transform (GRT) extends the original idea of the classical Radon transform introduced by [32] from integration over hyperplanes of $\mathbb{R}^d$ to integration over hypersurfaces, *i.e.* $(d-1)$-dimensional manifolds [37, 40, 41, 42, 43, 44]. The GRT has various applications, including Thermoacoustic Tomography, where the hypersurfaces are spheres, and Electrical Impedance Tomography, which requires integration over hyperbolic surfaces.

To formally define the GRT, we introduce a function $g$ defined on $X \times (\mathbb{R}^n \backslash \{0\})$ with $X \subset \mathbb{R}^d$. We say that $g$ is a *defining function* when it satisfies the four conditions below:

**H1.** *$g$ is a real-valued $C^\infty$ function on $X \times (\mathbb{R}^n \backslash \{0\})$*

**H2.** *$g(x, \theta)$ is homogeneous of degree one in $\theta$, i.e., $\forall \lambda \in \mathbb{R},\ g(x, \lambda\theta) = \lambda g(x, \theta)$.*

**H3.** *$g$ is non-degenerate in the sense that $\forall (x, \theta) \in X \times \mathbb{R}^n \backslash \{0\},\ \frac{\partial g}{\partial x}(x, \theta) \neq 0$.*

**H4.** *The mixed Hessian of $g$ is strictly positive, i.e. $det\left(\left(\frac{\partial^2 g}{\partial x^i \partial \theta^j}\right)_{i,j}\right) > 0$.*

Then, the GRT of $I \in L^1(\mathbb{R}^d)$ is the integration of $I$ over hypersurfaces characterized by the level sets of $g$, which are characterized by $H_{t,\theta} = \{x \in X \mid g(x, \theta) = t\}$.

Let $g$ be a defining function. The generalized Radon transform of $I$, denoted by $\mathcal{G}I$, is then formally defined as:

$$\mathcal{G}I(t, \theta) = \int_{\mathbb{R}^d} I(x)\delta(t - g(x, \theta))dx \tag{10}$$

Note that the standard Radon transform is a special case of the GRT with $g(x, \theta) = \langle x, \theta \rangle$. Figure 1 illustrates the slicing process for standard and generalized Radon transforms for the Half Moons dataset as input.

## 3.2 Generalized Sliced-Wasserstein and Max-Generalized Sliced-Wasserstein Distances

Following the definition of the SW distance in Equation (7), we define the generalized sliced $p$-Wasserstein distance using the generalized Radon transform as:

$$GSW_p(I_\mu, I_\nu) = \left( \int_{\Omega_\theta} W_p^p \big( \mathcal{G}I_\mu(\cdot, \theta), \mathcal{G}I_\nu(\cdot, \theta) \big) d\theta \right)^{\frac{1}{p}} \tag{11}$$

where $\Omega_\theta$ is a compact set of feasible parameters for $g(\cdot, \theta)$ (e.g., $\Omega_\theta = \mathbb{S}^{d-1}$ for $g(\cdot, \theta) = \langle \cdot, \theta \rangle$).

The GSW distance can also suffer from the projection complexity issue described before; that is why we formulate the maximum generalized sliced $p$-Wasserstein distance, which generalizes the max-SW distance as defined in (9):

$$\text{max-}GSW_p(I_\mu, I_\nu) = \max_{\theta \in \Omega_\theta} W_p \big( \mathcal{G}I_\mu(\cdot, \theta), \mathcal{G}I_\nu(\cdot, \theta) \big) \tag{12}$$

**Proposition 1.** *The generalized sliced $p$-Wasserstein distance and the maximum generalized sliced $p$-Wasserstein distance are, indeed, distances over $\mathcal{P}_p(\Omega)$ if and only if the generalized Radon transform is injective.*

The proof is given in the supplementary document.

**Remark 1.** *If the chosen generalized Radon transform is not injective, then we can only say that the GSW and max-GSW distances are pseudo-metrics: they still satisfy non-negativity, symmetry, the triangle inequality, and $GSW_p(I_\mu, I_\mu) = 0$ and $\text{max-}GSW_p(I_\mu, I_\mu) = 0$.*

**Remark 2.** *Proposition 1 shows that the injectivity of GRT is sufficient and necessary for GSW to be a metric. In this respect, our result brings a different perspective on the results of [23] by showing that SW is indeed distance since the standard Radon transform is injective.*

## 3.3 Injectivity of the Generalized Radon Transform

We have shown that the injectivity of the GRT is crucial for the GSW and max-GSW distances to be, indeed, distances between probability measures. Here, we enumerate some of the known defining functions that lead to injective GRTs.

The investigation of the sufficient and necessary conditions for showing the injectivity of GRTs is a long-standing topic [37, 44, 45, 41]. The circular defining function, $g(x, \theta) = \|x - r * \theta\|_2$ with $r \in \mathbb{R}^+$ and $\Omega_\theta = \mathbb{S}^{d-1}$ was shown to provide an injective GRT [43]. More interestingly, homogeneous polynomials with an odd degree also yield an injective GRT [46], *i.e.* $g(x, \theta) = \sum_{|\alpha|=m} \theta_\alpha x^\alpha$, where we use the multi-index notation $\alpha = (\alpha_1, \dots, \alpha_{d_\alpha}) \in \mathbb{N}^{d_\alpha}$, $|\alpha| = \sum_{i=1}^{d_\alpha} \alpha_i$, and $x^\alpha = \prod_{i=1}^{d_\alpha} x_i^{\alpha_i}$. Here, the summation iterates over all possible multi-indices $\alpha$, such that $|\alpha| = m$, where $m$ denotes the degree of the polynomial and $\theta_\alpha \in \mathbb{R}$. The parameter set for homogeneous polynomials is then set to $\Omega_\theta = \mathbb{S}^{d_\alpha - 1}$. We can observe that choosing $m = 1$ reduces to the linear case $\langle x, \theta \rangle$, since the set of the multi-indices with $|\alpha| = 1$ becomes $\{(\alpha_1, \dots, \alpha_d); \alpha_i = 1$ for a single $i \in [\![1, d]\!]$, and $\alpha_j = 0, \quad \forall j \neq i\}$ and contains $d$ elements.

While the polynomial projections form an interesting alternative to linear projections, their memory complexity $d_\alpha$ grows exponentially with the dimension of the data and the degree of the polynomial, hence deteriorate their potential in modern machine learning problems. As a remedy, given the current success of the neural networks in various application domains, a natural task in our context would be to come up with a neural network, which would yield a valid GSW or max-GSW, when used as the defining function in the GRT. As a neural network-based defining function, we propose a multi-layer fully connected network with 'leaky ReLU' activations. Under this specific network architecture, one can easily show that the corresponding defining function satisfies **H**1 to **H**4 on $(X \backslash \{0\}) \times (\mathbb{R}^n \backslash \{0\})$. On the other hand, it is highly non-trivial to show the injectivity of the associated GRT, therefore the GSW associated with this particular defining function is a pseudo-metric, as we discussed in Remark 1. However, as illustrated later on in Section 5, this neural network-based defining function still performs well in practice, and specifically, the non-differentiability of the leaky ReLU function at 0 does not seem to be a big issue in practice.

**Remark 3.** *With a neural network as the defining function, minimizing max-GSW between two distributions is analogical to adversarial learning, where the adversary network's goal is to distinguish the two distributions. In the max-GSW case, the adversary network (i.e. the defining function) seeks optimal parameters that maximize the GSW distance between the input distributions.*

# 4 Numerical Implementation

## 4.1 Generalized Radon Transforms of Empirical PDFs

In most machine learning applications, we do not have access to the distribution $I_\mu$ but to a set of samples $\{x_i\}_{i=1}^N$ drawn from $I_\mu$, for which the empirical density is: $I_\mu(x) \approx \frac{1}{N} \sum_{i=1}^N \delta(x - x_i)$ The GRT of the empirical density is then given by: $\mathcal{G}I_\mu(t, \theta) \approx \frac{1}{N} \sum_{i=1}^N \delta(t - g(x_i, \theta))$ Moreover, for high-dimensional problems, estimating $I_\mu$ in $\mathbb{R}^d$ requires a large number of samples. However, the projections of $I_\mu$, $\mathcal{G}I(\cdot, \theta)$, are one-dimensional and it may not be critical to have a large number of samples to estimate these one-dimensional densities.

## 4.2 Numerical Implementation of GSW Distances

Let $\{x_i\}_{i=1}^N$ and $\{y_j\}_{j=1}^N$ be samples respectively drawn from $I_\mu$ and $I_\nu$, and let $g(\cdot, \theta)$ be a defining function. Following the work of [30], the Wasserstein distance between one-dimensional distributions $\mathcal{G}I_\mu(\cdot, \theta)$ and $\mathcal{G}I_\nu(\cdot, \theta)$ can be calculated from sorting their samples and calculating the $L_p$ distance between the sorted samples. In other words, the GSW distance between $I_\mu$ and $I_\nu$ can be approximated from their samples as follows:

$$GSW_p(I_\mu, I_\nu) \approx \left( \frac{1}{L} \sum_{l=1}^L \sum_{n=1}^N |g(x_{i[n]}, \theta_l) - g(y_{j[n]}, \theta_l)|^p \right)^{1/p}$$

where $i[m]$ and $j[n]$ are the indices of sorted $\{g(x_i, \theta)\}_{i=1}^N$ and $\{g(y_j, \theta)\}_{j=1}^N$. The procedure to approximate the GSW distance is summarized the supplementary document.

## 4.3 Numerical Implementation of max-GSW Distances

To compute the max-GSW distance, we perform an EM-like optimization scheme: (a) for a fixed $\theta$, $g(x_i, \theta)$ and $g(y_i, \theta)$ are sorted to compute $W_p$, (b) $\theta$ is updated with a Projected Gradient Descent (PGD) step:

$$\theta = \underset{\Omega_\theta}{Proj}\left( Optim\left( \nabla_\theta \left( \frac{1}{N} \sum_{n=1}^N |g(x_{i[n]}, \theta) - g(y_{j[n]}, \theta)|^p \right), \theta \right) \right)$$

where $Optim(\cdot)$ refers to the preferred optimizer, for instance Gradient Descent (GD) or ADAM [47], and $\underset{\Omega_\theta}{Proj}(\cdot)$ is the operator projecting $\theta$ onto $\Omega_\theta$. For instance, when $\theta \in \mathbb{S}^{n-1}$, $\underset{\Omega_\theta}{Proj}(\theta) = \frac{\theta}{\|\theta\|}$.

**Remark 4.** *Here, we find the optimal $\theta$ by optimizing the actual $W_p$, as opposed to the heuristic approaches proposed in [28] and [30], where the pseudo-optimal slice is found via perceptrons or penalized linear discriminant analysis [48].*

Finally, once convergence is reached, the max-GSW distance is approximated with:

$$\text{max-}GSW_p(I_\mu, I_\nu) \approx \left( \frac{1}{N} \sum_{n=1}^N |g(x_{i[n]}, \theta^*) - g(y_{j[n]}, \theta^*)|^p \right)^{\frac{1}{p}}$$

The whole procedure is summarized as pseudocode in the supplementary document.

# 5 Experiments

In this section, we conduct experiments on the generalized Sliced-Wasserstein flows. We also implemented GSW-based auto-encoders, whose results are reported in the supplementary document due to space limitations. We provide the source code to reproduce the experiments of this paper.[2]

Our goal is to demonstrate the effects of the choice of the GSW distance in its purest form by considering the following problem: $\min_\mu GSW_p(\mu, \nu)$, where $\nu$ is a target distribution and $\mu$ is the source distribution, which is initialized to be the normal distribution. The optimization is then solved iteratively via: $\partial_t \mu_t = -\nabla GSW_p(\mu_t, \nu)$, $\mu_0 = \mathcal{N}(0, 1)$.

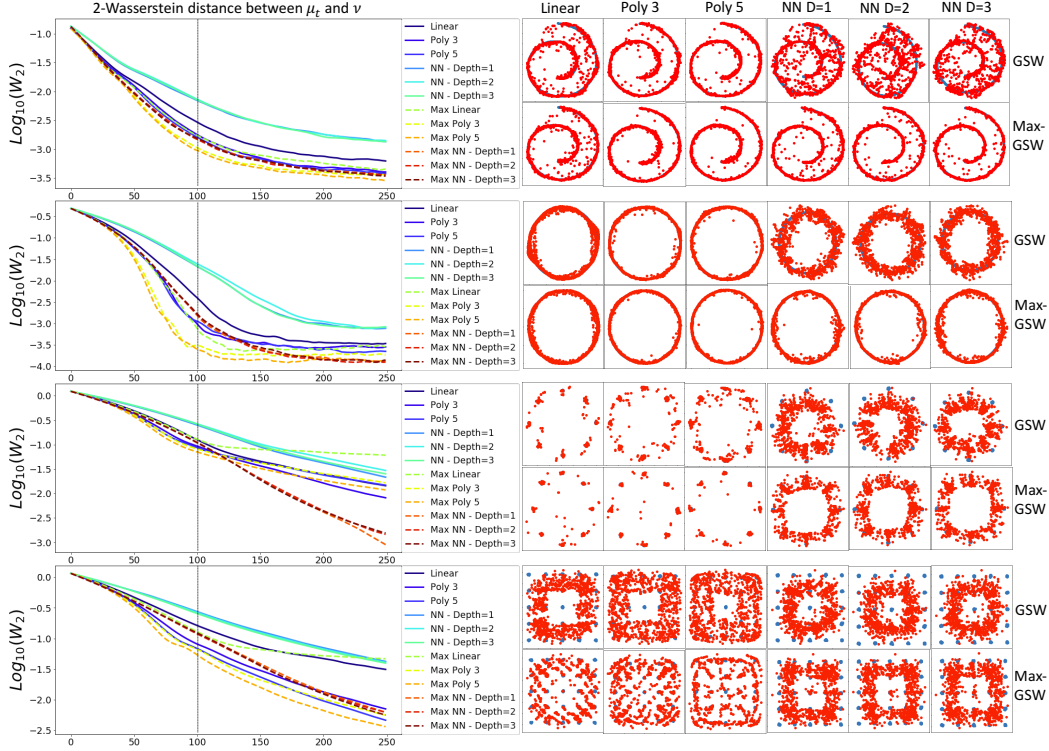

Figure 2: Log 2-Wasserstein distance between the source and target distributions as a function of the number of iterations for 4 classical target distributions.

We started by using 4 well-known distributions as the target, namely the 25-Gaussians, 8-Gaussians, Swiss Roll, and Circle distributions. We compare GSW and max-GSW for optimizing the flow with linear (*i.e.*, SW distance), homogeneous polynomials of degree 3 and 5, and neural networks with 1, 2, and 3 hidden layers as

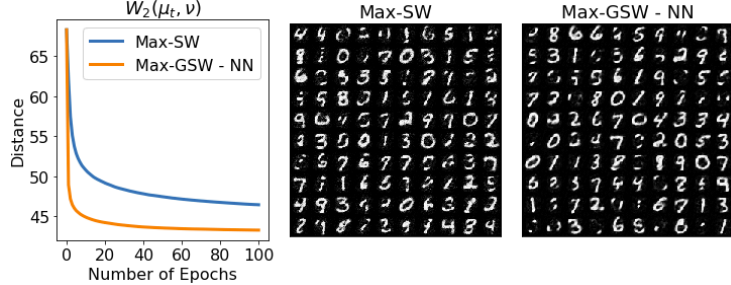

Figure 3: 2-Wasserstein distance between source and target distributions for the MNIST dataset.

defining functions. We used the exact same optimization scheme for all methods, and kept only $L = 1$ projection, and calculated the 2-Wasserstein distance between $\mu_t$ and $\nu$ at each iteration of the optimization (via solving a linear programming at each step). We repeated each experiment 100 times and reported the mean of the 2-Wasserstein distance for all target datasets in Figure 2. We also showed a snapshot of $\mu_t$ and $\nu$ at $t = 100$ iterations for all datasets. We observe that (i) max-GSW outperforms GSW, of course at the cost of an additional optimization, and (ii) while the choice of the defining function $g(\cdot, \theta)$ is data-dependent, one can see that the homogeneous polynomials are often among the top performers for all datasets. Specifically, SW is always outperformed by GSW with polynomial projections ('Poly 3' and 'Poly 5' in Figure 2, left) and by all the variants of max-GSW. Besides, max-linear-SW is consistently outperformed by max-GSW-NN. The only variant of GSW that is outperformed by SW is GSW with neural network-based defining function, which was expected because of its inherent complexity of approximating the integral over a very large domain (11) with a simple Monte Carlo average. To circumvent this issue, max-GSW replaces sampling with optimization.

To move to more realistic datasets, we considered GSW flows for the hand-written digit recognition dataset, MNIST, where we initialize 100 random images and optimize the flow via max-SW and max-

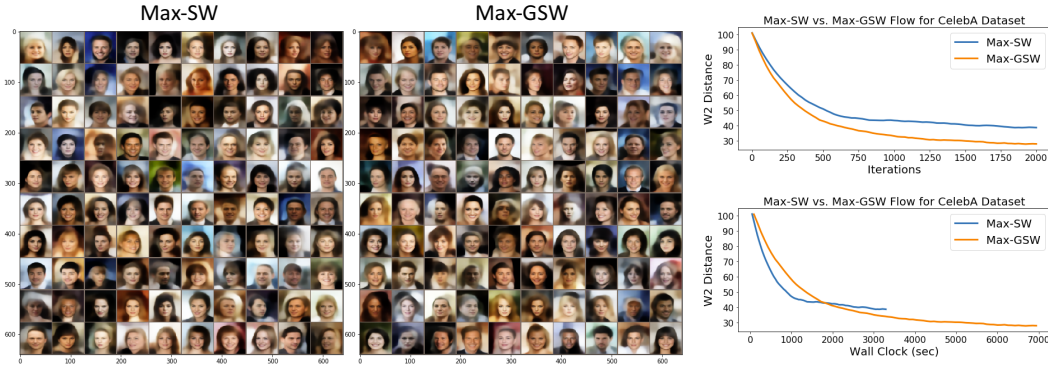

Figure 4: Flow minimization comparison between max-SW and max-GSW on the CelebA dataset.

GSW and measure the 2-Wasserstein distance between the $\mu_t$ (the 100 images) and $\nu$ (the training set of MNIST). See supplementary material for videos. Given the high-dimensional nature of the problem (i.e., 784-dims.) we cannot use the homogeneous polynomials due to memory constraints caused by the combinatorial growth of the coefficients. Therefore, we chose a 3-layer neural network for our defining function. Figure 3 shows the 2-Wasserstein between the source and target distributions as a function of number of training epochs. We observe that with the proposed approach the error is decreasing significantly faster when compared to the linear projections. We also observe this in the quality of the generated images, where we obtain crisper results.

Finally, we applied our methodology on a larger dataset, namely CelebA [49]. We performed flow optimization in a 256-dimensional latent space of a pre-trained auto-encoder, and compared max-SW with max-GSW using a 3 layer neural network. We then measured the 2-Wasserstein between the real and optimized distributions in the 256-dimensional latent space. Figure 4 shows the results of this experiment. As can be seen, max-GSW finds a better solution than max-SW in fewer iterations and the quality of the generated images is slightly better.

## 6  Conclusion

We introduced a new family of optimal transport metrics for probability measures that generalizes the sliced-Wasserstein distance: while the latter is based on linear slicing of distributions, we propose to perform nonlinear slicing. We provided theoretical conditions that yield the generalized sliced-Wasserstein distance to be, indeed, a distance function, and we empirically demonstrated the superior performance of the GSW and max-GSW distances over the classical sliced-Wasserstein distance in various generative modeling applications.

**Acknowledgements**

This work was partially supported by the United States Air Force and DARPA under Contract No. FA8750-18-C-0103. Any opinions, findings and conclusions or recommendations expressed in this material are those of the author(s) and do not necessarily reflect the views of the United States Air Force and DARPA. This work is also partly supported by the French National Research Agency (ANR) as a part of the FBIMATRIX project (ANR-16-CE23-0014) and by the industrial chair Machine Learning for Big Data from Télécom ParisTech.

## Footnotes

*Denotes equal contribution.

[2]See https://github.com/kimiandj/gsw.

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
