[Supplementary Material]

# Generalized Sliced Wasserstein Distances
## SUPPLEMENTARY DOCUMENT

**Soheil Kolouri**[1]*, **Kimia Nadjahi**[2]*, **Umut Şimşekli**[2,3], **Roland Badeau**[2], **Gustavo K. Rohde**[4]
1: HRL Laboratories, LLC., Malibu, CA, USA, 90265
2: LTCI, Télécom Paris, Institut Polytechnique de Paris, France
3: Department of Statistics, University of Oxford, UK
4: University of Virginia, Charlottesville, VA, USA, 22904
skolouri@hrl.com, gustavo@virginia.edu
{kimia.nadjahi, umut.simsekli, roland.badeau}@telecom-paris.fr

This document provides additional material to the main paper called Generalized Sliced-Wasserstein Distances.

## 1 Algorithm Pseudocodes

In Algorithms 1 and 2, we provide pseudocodes for the overall algorithm.

## 2 Non-negativity and Symmetry of the GSW and max-GSW Distances

We prove that the GSW and max-GSW distances satisfy non-negativity and symmetry, using the fact that the $p$-Wasserstein distance is known to be a proper distance function [1]. Let $\mu$ and $\nu$ be in $\mathcal{P}_p(\Omega)$.

### 2.1 Non-negativity

We use the non-negativity of the $p$-Wasserstein distance, *i.e.* $W_p(\mu, \nu) \geq 0$ for any $\mu$, $\nu$ in $\mathcal{P}_p(\Omega)$, to show that the GSW and max-GSW distances are non-negative as well:

$$GSW_p(I_\mu, I_\nu) = \left( \int_{\Omega_\theta} W_p^p \big( \mathcal{G}I_\mu(.,\theta), \mathcal{G}I_\nu(.,\theta) \big) d\theta \right)^{\frac{1}{p}}$$
$$\geq \left( \int_{\Omega_\theta} (0)^p d\theta \right)^{\frac{1}{p}} = 0$$

$$\text{max-}GSW_p(I_\mu, I_\nu) = \max_{\theta \in \Omega_\theta} W_p \big( \mathcal{G}I_\mu(\cdot,\theta), \mathcal{G}I_\nu(\cdot,\theta) \big)$$
$$= W_p \big( \mathcal{G}I_\mu(\cdot,\theta^*), \mathcal{G}I_\nu(\cdot,\theta^*) \big)$$
$$\geq 0$$

where $\theta^* = \arg\max_{\theta \in \Omega_\theta} W_p(\mathcal{G}I_\mu(\cdot,\theta), \mathcal{G}I_\nu(\cdot,\theta))$.

### 2.2 Symmetry

Since the $p$-Wasserstein distance is symmetric, we have $W_p(\mu, \nu) = W_p(\nu, \mu)$.

**Algorithm 1** GSW Distance

---

**input** $\{x_i \sim I_\mu\}_{i=1}^N$, $\{y_i \sim I_\nu\}_{i=1}^N$, order $p$,
        number of slices $L$, defining function $g$
  Initialize $d = 0$
  **for** $l = 1$ to $L$ **do**
     Sample $\theta_l$ from $\Omega_\theta$ uniformly
     Compute $\hat{x}_i = g(x_i, \theta_l)$ and $\hat{y}_i = g(y_i, \theta_l)$ for each $i$
     Sort $\hat{x}_i$ and $\hat{y}_j$ in ascending order s.t. $\hat{x}_{i[n]} \leq \hat{x}_{i[n+1]}$ and $\hat{y}_{j[n]} \leq \hat{y}_{j[n+1]}$
     $d = d + \frac{1}{L}\sum_{n=1}^N |\hat{x}_{i[n]} - \hat{y}_{i[n]}|^p$
  **end for**
**output** $d^{\frac{1}{p}} \approx GSW_p(I_\mu, I_\nu)$

---

---

**Algorithm 2** Max-GSW Distance

---

**input** $\{x_i \sim I_\mu\}_{i=1}^N$, $\{y_j \sim I_\nu\}_{j=1}^N$,
        order $p$, defining function $g(x, \theta)$
  Randomly initialize $\theta \in \Omega_\theta$
  **while** $\theta$ has not converged **do**
     Compute $\hat{x}_i = g(x_i, \theta_l)$ and $\hat{y}_i = g(y_i, \theta_l)$ for each $i$
     Sort $\hat{x}_i$ and $\hat{y}_j$ in ascending order s.t. $\hat{x}_{i[n]} \leq \hat{x}_{i[n+1]}$ and $\hat{y}_{j[n]} \leq \hat{y}_{j[n+1]}$
     $\theta = \underset{\Omega_\theta}{Proj}(Optim(\nabla_\theta(\frac{1}{N}\sum_{n=1}^N |\hat{x}_{i[n]} - \hat{y}_{j[n]}|^p), \theta))$
  **end while**
  Sort $\hat{x}_i$ and $\hat{y}_i$ in ascending order
  $d = \frac{1}{N}\sum_{n=1}^N |\hat{x}_{i[n]} - \hat{y}_{i[n]}|^p$
**output** $d^{\frac{1}{p}} \approx$ max-$GSW_p(I_\mu, I_\nu)$

---

In particular, we can write for all $\theta \in \Omega_\theta$:

$$W_p(\mathcal{G}I_\mu(\cdot, \theta), \mathcal{G}I_\nu(\cdot, \theta)) = W_p(\mathcal{G}I_\nu(\cdot, \theta), \mathcal{G}I_\mu(\cdot, \theta)),\qquad(1)$$

$$\max_{\theta \in \Omega_\theta} W_p(\mathcal{G}I_\mu(\cdot, \theta), \mathcal{G}I_\nu(\cdot, \theta)) = \max_{\theta \in \Omega_\theta} W_p(\mathcal{G}I_\nu(\cdot, \theta), \mathcal{G}I_\mu(\cdot, \theta))\qquad(2)$$

The symmetry of the GSW and max-GSW distances follows from Equations (1) and (2) respectively.

# 3 Proof of Proposition 1

*Proof.* The non-negativity and symmetry are direct consequences of the fact that the Wasserstein distance is a metric [1]: see the previous sections.

We prove the triangle inequality for $GSW_p$ and max-$GSW_p$. Let $\mu_1$, $\mu_2$ and $\mu_3$ in $\mathcal{P}_p(\Omega)$. Since the Wasserstein distance satisfies the triangle inequality, we have, for all $\theta \in \Omega_\theta$,

$$W_p(\mathcal{G}\mathcal{I}_{\mu_1}(\cdot, \theta), \mathcal{G}\mathcal{I}_{\mu_3}(\cdot, \theta)) \leq W_p(\mathcal{G}\mathcal{I}_{\mu_1}(\cdot, \theta), \mathcal{G}\mathcal{I}_{\mu_2}(\cdot, \theta)) \\ + W_p(\mathcal{G}\mathcal{I}_{\mu_2}(\cdot, \theta), \mathcal{G}\mathcal{I}_{\mu_3}(\cdot, \theta))$$

Therefore, we can write:

$$GSW_p(I_{\mu_1}, I_{\mu_3}) = \left( \int_{\Omega_\theta} W_p^p(\mathcal{G}I_{\mu_1}(\cdot, \theta), \mathcal{G}I_{\mu_3}(\cdot, \theta)) d\theta \right)^{\frac{1}{p}}$$

$$\leq \left( \int_{\Omega_\theta} \left( W_p(\mathcal{G}I_{\mu_1}(\cdot, \theta), \mathcal{G}I_{\mu_2}(\cdot, \theta)) \right. \right.$$

$$\left. \left. + W_p(\mathcal{G}I_{\mu_2}(\cdot, \theta), \mathcal{G}I_{\mu_3}(\cdot, \theta)) \right)^p d\theta \right)^{\frac{1}{p}}$$

$$\leq \left( \int_{\Omega_\theta} W_p^p(\mathcal{G}I_{\mu_1}(\cdot, \theta), \mathcal{G}I_{\mu_2}(\cdot, \theta)) d\theta \right)^{\frac{1}{p}}$$

$$+ \left( \int_{\Omega_\theta} W_p^p(\mathcal{G}I_{\mu_2}(\cdot, \theta), \mathcal{G}I_{\mu_3}(\cdot, \theta)) d\theta \right)^{\frac{1}{p}} \tag{3}$$

where inequality (3) follows from the application of the Minkowski inequality in $L^p(\Omega_\theta)$. We conclude that $GSW_p$ satisfies the triangle inequality.

Let $\theta^* = \arg\max_{\theta \in \Omega_\theta} W_p(\mathcal{GI}_{\mu_1}(\cdot, \theta), \mathcal{GI}_{\mu_3}(\cdot, \theta))$; then,

$$\text{max-}GSW_p(I_{\mu_1}, I_{\mu_3}) = \max_{\theta \in \Omega_\theta} W_p(\mathcal{GI}_{\mu_1}(\cdot, \theta), \mathcal{GI}_{\mu_3}(\cdot, \theta))$$

$$= W_p(\mathcal{GI}_{\mu_1}(\cdot, \theta^*), \mathcal{GI}_{\mu_3}(\cdot, \theta^*))$$

$$\leq W_p(\mathcal{GI}_{\mu_1}(\cdot, \theta^*), \mathcal{GI}_{\mu_2}(\cdot, \theta^*))$$

$$+ W_p(\mathcal{GI}_{\mu_2}(\cdot, \theta^*), \mathcal{GI}_{\mu_3}(\cdot, \theta^*))$$

$$\leq \max_{\theta \in \Omega_\theta} W_p(\mathcal{GI}_{\mu_1}(\cdot, \theta), \mathcal{GI}_{\mu_2}(\cdot, \theta))$$

$$+ \max_{\theta \in \Omega_\theta} W_p(\mathcal{GI}_{\mu_2}(\cdot, \theta), \mathcal{GI}_{\mu_3}(\cdot, \theta))$$

$$\leq \text{max-}GSW_p(I_{\mu_1}, I_{\mu_2}) + \text{max-}GSW_p(I_{\mu_2}, I_{\mu_3})$$

So max-$GSW_p$ also satisfies the triangle inequality.

Since $W_p(\mu, \mu) = 0$ for any $\mu$, we have $GSW_p(I_\mu, I_\nu) = 0$ and max-$GSW_p(I_\mu, I_\nu) = 0$. Now, $GSW_p(I_\mu, I_\nu) = 0$ or max-$GSW_p(I_\mu, I_\nu) = 0$ is equivalent to $\mathcal{G}I_\mu(\cdot, \theta) = \mathcal{G}I_\nu(\cdot, \theta)$ for almost all $\theta \in \Omega_\theta$. Therefore, GSW and max-GSW are distances if and only if $\mathcal{G}I_\mu(\cdot, \theta) = \mathcal{G}I_\nu(\cdot, \theta)$ implies $\mu = \nu$, *i.e.* the GRT is injective. □

## 4 Implementation Details

The PyTorch [2] implementation of our paper is available here[2]. Here we clarify some of the implementation details used in our paper. First, the 'critic iteration' for the adversarial training, and the projection maximization for the max-GSW distances, were set to be equal to $50$. For all optimizations, we used ADAM [3] optimizer with learning rate $lr = 0.001$ and PyTorch's default momentum parameters.

We used $3 \times 3$ convolutional filters in both encoder and decoder architectures. Encoder architecture:

$$
\begin{aligned}
x \in \mathbb{R}^{28 \times 28} \quad &\rightarrow \quad Conv_{16} \rightarrow LeakyReLU_{0.2} \\
&\rightarrow \quad Conv_{16} \rightarrow LeakyReLU_{0.2} \\
&\rightarrow \quad AvgPool_2 \\
&\rightarrow \quad Conv_{32} \rightarrow LeakyReLU_{0.2} \\
&\rightarrow \quad Conv_{32} \rightarrow LeakyReLU_{0.2} \\
&\rightarrow \quad AvgPool_2 \\
&\rightarrow \quad Conv_{64} \rightarrow LeakyReLU_{0.2} \\
&\rightarrow \quad Conv_{64} \rightarrow LeakyReLU_{0.2} \\
&\rightarrow \quad AvgPool_2 \rightarrow Flatten \\
&\rightarrow \quad FC_{128} \rightarrow LeakyReLU_{0.2} \\
&\rightarrow \quad FC_2
\end{aligned}
$$

Decoder architecture:

$$
\begin{aligned}
z \in \mathbb{R}^2 \quad &\rightarrow \quad FC_{128} \rightarrow LeakyReLU_{0.2} \\
&\rightarrow \quad FC_{1024} \rightarrow LeakyReLU_{0.2} \\
&\rightarrow \quad Reshape(4 \times 4 \times 64) \rightarrow Upsample_2 \\
&\rightarrow \quad Conv_{64} \rightarrow LeakyReLU_{0.2} \\
&\rightarrow \quad Conv_{64} \rightarrow LeakyReLU_{0.2} \\
&\rightarrow \quad Upsample_2 \\
&\rightarrow \quad Conv_{32} \rightarrow LeakyReLU_{0.2} \\
&\rightarrow \quad Conv_{32} \rightarrow LeakyReLU_{0.2} \\
&\rightarrow \quad Upsample_2 \\
&\rightarrow \quad Conv_{16} \rightarrow LeakyReLU_{0.2} \\
&\rightarrow \quad Conv_1
\end{aligned}
$$

## 5 Generative Modeling via Auto-Encoders

We now demonstrate the application of the GSW and max-GSW distances in generative modeling. We specifically use the recently proposed Sliced-Wasserstein Auto-Encoder (SWAE) [4] framework, which penalizes the distribution of the encoded data in the latent space of the auto-encoder to follow a prior samplable distribution, $p_Z$. More precisely, let $\{x_n \sim p_X\}_{n=1}^N$ be i.i.d. samples from $p_X$, $\phi(x, \gamma_\phi) : \mathcal{X} \to \mathcal{Z}$ and $\psi(z, \gamma_\psi) : \mathcal{Z} \to \mathcal{X}$ be the parametric encoder and decoder (e.g., CNNs) with parameters $\gamma_\phi$ and $\gamma_\psi$, respectively. Then SWAE's objective function [4] is defined as:

$$
\min_{\gamma_\phi, \gamma_\psi} \mathbb{E}_x[c(x, \psi(\phi(x, \gamma_\phi), \gamma_\psi))] + \lambda SW(p_{\phi(x,\gamma_\phi)}, p_Z) \tag{4}
$$

where $\lambda$ is the regularizer coefficient for matching the encoded distribution to $p_Z$. Here, we substitute the SW distance in Equation (4) with GSW and max-GSW distances. Specifically, we encode the MNIST dataset [5] into the encoder's latent space and enforce the distribution of the embedded data to follow a specific prior distribution, e.g. the Swiss Roll distribution as shown in Figure 1, while we simultaneously enforce the encoded features to be decodable to the original input images. Since the latent dimensionality is small in this case, we can apply the polynomial defining functions, without needing to apply the neural network-based one.

We ran the optimization in Equation (4) with GSW distances, which we denote as GSWAE, with linear, polynomial degree 3, and polynomial degree 5 and their max versions. The results are shown in Figure 2.

Figure 1: The SWAE architecture. The embedded data in the latent space is enforced to follow a prior samplable distribution $p_Z$.

Figure 2: Results on GSWAE, with linear (i.e., SWAE), polynomial degree 3 and polynomial degree 5 defining functions and the corresponding max-GSWAE results (The results are shown after 10 epochs on MNIST).

## Footnotes

*Denotes equal contribution.

[2]https://github.com/.../GSW/