[Reviews · NeurIPS 2019]

Reviewer 1



Authors define generalized sliced wasserstein distances by appealing to the notion of generalized radon transform. They provide some elementary properties (e.g. saying when it is a distance),argue they are preferred over vanilla sliced wasserstein and conclude with some empirical validation. Clarity: the paper is clear and well written Originality: the material presented here is original and has not appeared elsewhere Quality and significance: here are my concerns, which altogether prevent me to recommend publication. I recognize the hard work that has been put on this paper, and I think it is a nice idea, but I believe the results should be strengthen. My main concern is that it is not made clear why the generalized sliced distance is a significant improvement over the vanilla sliced distance. It is hard to tell from current experiments: figure 2 compares between several generalized sliced and the linear one. In many cases results are better for the linear distance, making me wonder whether the improvement seen in some cases might be an artifact of multiple comparisons. Because of this, I am not fully convinced that the improvement on figure 3 is not a consequence of having chosen the best model. Authors should therefore make a strong point about e.g. how to tune the generalized distance in such a way that it will consistently outperforms the linear one (and not overfitting). A discussion in terms of complexity is also encouraged Currently, the first 5 pages of the paper are mostly definitions and review of other results, but I would expect more substance for a neurips submission there. If that section is shrunk a bit and replaced with more experimental validation (some of this is available in the supplement, but still needs some polishing) it would mean a substantial improvement.

Reviewer 2



The paper establishes connection of sliced-Wasserstein distance to Randon transform. From this viewpoint, a family of generalized SW distances are defined from generalized Randon transform. This enables a number of contributions including in particular non-linear slicing, and using only a single projection. Empirical evaluations shows the advantages of the proposed distance family over the classical sliced-Wasserstein distance such as much less iterations and better data generation quality. The problem is important. The analysis is new and convincing. Empirical evaluations also nicely demonstrate the proposed generalized distance scheme.

Reviewer 3



***** After Author Response and Reviewer Discussions ***** I have gone through all the other reviews, the meta-reviewer's comment, and the authors' feedback. I will keep my evaluation unchanged, and emphasize a point found by the meta-reviewer, that is the authors should not claim ReLU or leaky ReLU to satisfy the smooth conditions. **************************************************************** Originality: The methodology provided in this paper is, to our best knowledge, new. It is an innovative combination of the well-known techniques, namely the SW and Radon Transform. The paper is clearly different from the previous contributions and the related literature is sufficiently cited. Quality: The submission is technically sound. The claims are well supported by the analysis, although due to time restriction I have not gone through the proofs. This work is complete. The authors are honest about evaluating both the strengths and weakness of their work. Clarity: The submission is clearly and elegantly organized and written with good style. Since the proof is provided in the supplementary materials, it is sufficient for the expert readers to validate the claims. Significance: The results are obviously important. I believe that the method developed here would attract lots of interests and push the frontier of the research forward. For the suitable problems, the methodology would advance the state of the art in a demonstratable way. This paper provides unique theoretical approach.

[Author Response · NeurIPS 2019]

We thank the reviewers for their feedback and time. We are happy to see that they found our article clear, well-written,
original, innovative, and technically sound. The detailed responses are given below.

**R1:** We thank the reviewer for their valuable comments. We believe that we have now addressed all the raised issues.
We hope this would help the reviewer to reconsider their score.

*"... results are better for the linear distance, making me wonder whether the improvement ... might be an artifact..."*
We suspect that there has been a simple misunderstanding, which we first would like to clarify. As opposed to the
claim, Fig 2 actually shows that linear SW is *always* outperformed by GSW with *polynomial* projections ('Poly 3' and
'Poly 5' in the legend) and also by all the variants of max-GSW (dashed lines). Besides, the figure also shows that
*max-linear-SW* is consistently outperformed by *max-GSW-NN* as well. Then, if we consider Fig 3, we observe a clear
continuation of this behavior, where max-GSW-NN outperforms max-SW on a real-data experiment.

Therefore, we would like to underline that the performance gain in Fig 3 is not due to choosing the best model setting
or parameter tweaking. Besides, we would like to indicate that, we have provided our code (to obtain Fig 2) in the
supplementary material, and the same code can be easily adapted to reproduce Fig 3, by simply loading MNIST in
place of generating synthetic data. We invite the reviewer to use our code to verify the reproducibility.

Again regarding Fig 2, we would also like to clarify that, the only variant of GSW that is outperformed by linear
SW is the GSW with neural network-based defining function (which is different from *max*-GSW-NN). This variant
is unsurprisingly not performing well due to its inherent complexity of approximating the integral over a very large
domain (Eq. 11) with a simple Monte Carlo average. On the contrary, max-GSW circumvents this issue by replacing
sampling with optimization (note that for the same reason, [35] also preferred optimization over sampling). We agree
that we could have emphasized more this point in the paper and we will add a short discussion to be more clear about it.

*Experiments.* Up on the reviewer's sugges-
tion, we have contrasted our experiments
with the ones of the three mentioned papers.
All these papers first consider an experiment
on synthetic data and then apply their ap-
proaches on real data. While we have the
exact same structure in our experiments, we
noticed that the main difference between our
experiments and theirs is the application on a

**Fig. R1.** Results on CelebA

larger dataset. We have now conducted additional experiments on the CelebA dataset (Fig R1). The results are inline
with our existing results: our approach finds a better solution in less number of iterations and the quality of the generated
images is slightly better. We will add these new results to the paper. On the other hand, we have comparisons to SW
and max-SW, which have already been compared to other baseline methods in [28,34]. Therefore, with the inclusion of
these new results, we believe that our empirical validation is as comprehensive as the ones of the mentioned papers.

*Computational/statistical complexities.* We will add a paragraph about the com-
putational requirements as requested. Regarding the statistical complexity, we
can consider the sample complexity of GSW. This has been established very
recently for SW and max-SW only for Gaussian measures [34], and proving anal-
ogous results for GSW is out of our scope. However, to gain intuition about the
sample complexity of GSW, we computed max-GSW and max-SW for varying
number of samples drawn from two different Swiss Roll distributions (Fig R2).
The distances exhibit the same behavior: we conjecture that GSW and SW have
similar sample complexities. We will discuss this point in the supp. document.

**Fig. R2.** Sample complexity

*Preliminaries.* We indeed deliberately allocated some space to introduce pre-
liminary notions that might not be familiar to the ML community: the Radon
transform and its generalization, which are the key constituents of GSW, are
mostly used in tomography. Nevertheless, we will shorten Sec.2 as suggested
and move some of the experiments of the supp. document as well as the ones
presented here. In the end, our article will contain results on *several synthetic*
*datasets, MNIST and CelebA, using GSW-flows and GSW-auto-encoders.* We believe this meets the NeurIPS standards.

**R2:** We thank the reviewer for the positive evaluation. We have actually provided the code to reproduce Fig 2: see the
supplementary material of the submission. The code for the other experiments is very similar and will also be publicly
released.

**R3:** We are grateful to the reviewer for their highly positive and encouraging feedback.

[Meta-Review · NeurIPS 2019]

The reviewers found the paper original and found it deserve an accept. Nevertheless some concern about the claims in 3.3 was raised during the discussion. The authors say that a "multi-layer fully connected network with ‘leaky ReLU’ activations" satisfies assumptions H1-H4. This is obviously false since leaky relu is non differentiable, hence at least H1 of C^\infty is false. This would means that their NN implementations do not fit in their theoretical framework (although it works very well in practice). The paper should be updated to correct any false claim in the final revision.